# Co-axial heterostructures integrating palladium/titanium dioxide with carbon nanotubes for efficient electrocatalytic hydrogen evolution

Giovanni Valenti[1,*], Alessandro Boni[1,*], Michele Melchionna[2], Matteo Cargnello[3], Lucia Nasi[4], Giovanni Bertoni[4], Raymond J. Gorte[5], Massimo Marcaccio[1], Stefania Rapino[1], Marcella Bonchio[6], Paolo Fornasiero[2,7], Maurizio Prato[2,8,9] & Francesco Paolucci[1,10]

Considering the depletion of fossil-fuel reserves and their negative environmental impact, new energy schemes must point towards alternative ecological processes. Efficient hydrogen evolution from water is one promising route towards a renewable energy economy and sustainable development. Here we show a tridimensional electrocatalytic interface, featuring a hierarchical, co-axial arrangement of a palladium/titanium dioxide layer on functionalized multi-walled carbon nanotubes. The resulting morphology leads to a merging of the conductive nanocarbon core with the active inorganic phase. A mechanistic synergy is envisioned by a cascade of catalytic events promoting water dissociation, hydride formation and hydrogen evolution. The nanohybrid exhibits a performance exceeding that of state-of-the-art electrocatalysts (turnover frequency of 15000 $H_2$ per hour at 50 mV overpotential). The Tafel slope of $\sim 130$ mV per decade points to a rate-determining step comprised of water dissociation and formation of hydride. Comparative activities of the isolated components or their physical mixtures demonstrate that the good performance evolves from the synergistic hierarchical structure.

[1] Department of Chemistry 'Giacomo Ciamician', University of Bologna and INSTM, via Selmi 2, Bologna 40126, Italy. [2] Department of Chemical and Pharmaceutical Sciences and INSTM, University of Trieste, via L. Giorgieri 1, Trieste 34127, Italy. [3] Department of Chemical Engineering and SUNCAT Center for Interface Science and Catalysis, Stanford University, Stanford, California 94305, USA. [4] IMEM-CNR Institute, Parco area delle Scienze 37/A, Parma 43124, Italy. [5] Department of Chemical and Biomolecular Engineering, University of Pennsylvania, 220 S. 33rd Street, Philadelphia, Pennsylvania 19104, USA. [6] Department of Chemical Sciences and ITM-CNR, University of Padova, via F. Marzolo 1, Padova 35131, Italy. [7] ICCOM-CNR Trieste Associate Unit, University of Trieste, via L. Giorgieri 1, Trieste 34127, Italy. [8] Nanobiotechnology Laboratory, CIC biomaGUNE, Paseo de Miramón 182, Donostia-San Sebastián 20009, Spain. [9] Ikerbasque, Basque Foundation for Science, Bilbao 48013, Spain. [10] ICMATE-CNR Bologna Associate Unit, University of Bologna, via Selmi 2, Bologna 40126, Italy. * These authors contributed equally to this work. Correspondence and requests for materials should be addressed to P.F. (email: pfornasiero@units.it) or to M.P. (email: prato@units.it) or to F.P. (email: francesco.paolucci@unibo.it).

Renewable energy running on hydrogen embraces a broad range of current technologies so that the hydrogen transition offers a pathway towards the sustainable goal of a carbon-neutral economy. On one side, hydrogen generated from water electrolysis can be linked to renewable electricity (solar, wind and geothermal) to bio-driven devices (microbial cells and algae plants) and to artificial photosynthesis schemes[1–3]. On the other end, hydrogen, as carbon-neutral energy vector, is pivotal for fuel cell technology, as primary fuel for generators and engines, and as chemical commodity for some major industrial processes for synthetic fertilizer or hydrocarbon productions. Furthermore, in situ $H_2$ production provides a greener and safer approach to drive key selective transformations[4], including $CO_2$ reduction for environmental remediation. Therefore, increasing the efficiency of $H_2$ production using water as the primary source, while offering a broad flexibility of production and delivery conditions, is mandatory to allow a transition to hydrogen-based energy schemes. The turning point is to be sought within the hydrogen evolution reaction (HER), occurring by the reduction of water at the electrolyser cathode. The HER efficiency and tunability is mainly dependent on the electrocatalyst package in terms of its composition and morphology. Pt-based cathodes are generally considered as the best choice for HER electrocatalysis, by virtue of a generally low overpotential, high turnover frequency and wide pH range of application[3,5–7]. However, Pt-based catalysts are plagued by the unfavourable market price, together with their facile poisoning by water contaminants and the marked loss of activity at under neutral and alkaline conditions[8]. Research has therefore turned to Pt-free alternatives, and oxides[9], sulfides[10,11], nitrides[12,13], and phosphides[2,14–16], either as pure phases or in combination with metals, are being investigated for this purpose under several different conditions[17,18]. An open problem is still the engineering of innovative catalytic cathodes to tackle efficient HER at neutral pH, as this is required to match the operating conditions of some highly promising technologies such as the microbial electrolysis cells or solar water splitting for bio-inspired artificial photosynthesis[19]. In all cases, key steps for the overall process are as follows: (i) the separation of HER half-reaction (HER, $2H^+ + 2e^- = H_2$) from the anodic counterpart; (ii) a tailored design of the electrocatalytic interface to promote the formation of HER intermediates ($H_{ad}$) via water dissociation at neutral pH ($H_2O + e^- \rightarrow H_{ad} + OH^-$, Volmer step at pH = 7); and (iii) the correct trafficking of reducing and oxidizing species between these two compartments[20].

For this purpose, an attractive strategy can be envisaged by a hierarchical design of the catalyst nanoarchitecture[21], where selected components are arranged to leverage the expected mechanistic functions, namely: (i) water dissociation; (ii) protons reduction; and (iii) electrons translocation[22]. Carbon nanotubes (CNTs) are prominent materials because of their high surface areas, electrical conductivity and stability in acidic or basic aqueous solutions[23]. Their one-dimensional morphology is then especially suitable for electrochemical applications, as they behave like nanowires that can connect the active nanostructures to the electrodes. Indeed, many approaches have shown the benefits of CNTs for electrochemical hydrogen production at low overvoltages[24]. Furthermore, CNT nanoscaffolds can template the arrangement of the metal-based catalytic interfaces by enhancing the active surface area, while connecting the phase boundaries to improve the electrocatalytic efficiency of the process.

The combination of HER electrocatalyst with metal–oxide interfaces has been proposed to boost the hydrogen evolution coupled with water dissociation in the absence of acidic electrolytes. This idea has been demonstrated in the tailored design of Pt surfaces[8], and it is here applied to the concept of $Pd/TiO_2$-based units assembled in hierarchical nanostructures for platinum-free HER. This approach has the advantage to replace Pt with Pd, a more abundant metal[25], extensively used in automotive catalytic converters. Notably, the possible large scale diffusion of highly efficient hydrogen-based fuel cell technology and their application in the transportation technology would markedly reduce the worldwide demand of Pd in automotive converters, leaving open options for other applications, such as HER electrocatalysis. Finally, Pd has the additional value to be nontoxic or allergenic as for other investigated systems such as the Ni-based ones.

Building on these concepts, we prepare a hybrid HER catalyst, **f-MWCNTs@Pd/TiO₂**, based on the combination of functionalized multi-walled carbon nanotubes (f-MWCNTs) and Pd nanoparticles that were mutually integrated within a titanium dioxide ($TiO_2$) shell[26]. The resulting nanocomposite provides a hierarchical organization of the nanocarbon, metal and metal oxide interfaces, where the specific roles of the individual components are merged to direct HER with an exceptional proficiency. The CNT nanowires enable electron transfer from the electrode to the titania shell, so as to feed the catalytically active Pd nanoparticles. Furthermore, the hydrophilicity of the amorphous $TiO_2$ coating, with its attendant of OH groups, counteracts the CNT hydrophobic nature, and favours water adsorption/dissociation equilibria, and proton translocation steps. In particular, $TiO_2$ defects bordering the MWCNTs surface are expected to play a pivotal role in the HER mechanism. The final outcome is a high efficiency of HER at neutral pH for **f-MWCNTs@Pd/TiO₂** that reaches, at zero overpotential, a turnover frequency $(TOF)_0$, of 9,460 $H_2$ per hour. This result may set a benchmark for the HER efficiency, as state-of-the art heterogeneous electrocatalysts under similar conditions are an order of magnitude slower. By electrochemical investigations, the catalyst performance is ascribed to an increase in the doping levels of $TiO_2$, together with the generation of surface states at the interface between the semiconductor and the MWCNTs[27]. Extensive experimental results also demonstrate the presence of cooperative effects within the constituents of **f-MWCNTs@Pd/TiO₂**, whose HER activity greatly outperform that of the isolated components.

## Results

**Synthesis of the nanohybrids.** The first step of the synthesis of the MWCNT heterostructures integrating **Pd/TiO₂** requires the functionalization of the MWCNT surface to promote the metal–oxide adhesion (Fig. 1a). A twofold strategy was adopted to provide the nanocarbon surface with anchor groups, enabling the growth of a co-axial $TiO_2$ outer layer that envelopes the MWCNT throughout its length. The MWCNT were thus treated: (i) under oxidation conditions to yield **Ox-MWCNTs@Pd/TiO₂** or (ii) by a via diazonium salt radical addition according to the well-known Tour reaction protocol[28], yielding **Tour-MWCNTs@Pd/TiO₂** (Fig. 1a). These functionalized nanoscaffolds provide a direct insight into the role of the chemical-spacer (–OH; –COOH versus benzoic residues) separating the $TiO_2$ shell from the nanocarbon surface, while the final thermal annealing step turns out to seal the MWCNT/TiO₂ interface, thus overriding any structural difference, as addressed by the electrocatalytic screening (see below). The co-axial heterostructures self-assemble in solution by a simple one-pot protocol, where the f-MWCNTs are mixed within a Ti(OBu)x phase integrating the Pd nanoparticles (Fig. 1a). Further details on the synthesis can be found in the Supplementary Note 1.

**Structural characterization.** Structural and chemisorption evidences confirm that the Pd nanoparticles are deeply embedded

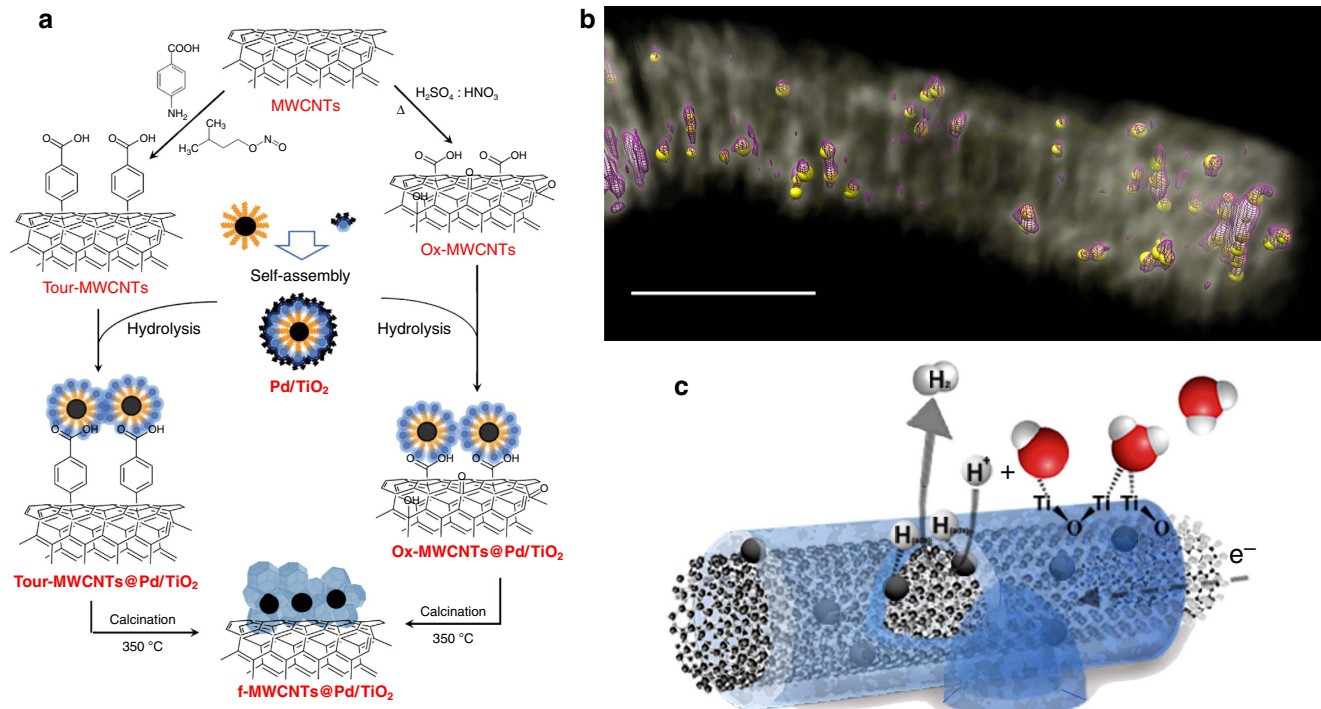

**Figure 1 | Nanostructured hydrogen-evolving cathode f-MWCNTs@Pd/TiO₂.** (**a**) Schematic representation of the synthetic protocols yielding the **f-MWCNTs@Pd/TiO₂** electrocatalysts. (**b**) STEM tomographic reconstruction of **f-MWCNTs@Pd/TiO₂**. Here the region of high density corresponding to the Pd particles is rendered with a violet mesh (raw data). The Pd particles are then localized by placing spherical markers centred in the regions of high density (yellow spheres). Scale bar, 100 nm. (**c**) General scheme for the HER proposed mechanism by **f-MWCNTs@Pd/TiO₂** at neutral pH. Water absorption and dissociation occurs at the hydrophilic, porous, TiO₂ outer layer (blue shell), promoting the formation of $H_{ad}$ (grey) intermediates on the catalytic Pd sites (black).

into a mesoporous network composed of nanometric TiO₂ particles, well connected with each other but leaving accessible the metal phase (Fig. 1b; Supplementary Figs 1 and 2)[29]. MWCNTs guarantee the electrical conductivity and the efficient delivery of electrons from the electrode to the titania layer. In turn, because of the favourable positions of the titania conduction band and Pd Fermi level, efficient electron transfer from the titania to the Pd occurs, providing the required electrons to the Pd nanoparticles to reduce protons to H₂ (Fig. 1c). The process is facilitated by the intimate three-dimensional contact between the Pd nanoparticles and TiO₂ that maximize the metal support interface.

Figure 2a–c shows representative transmission electron microscopy (TEM) images of the nanoarchitectures, where a uniform titania layer surrounding the f-MWCNTs is clearly observed. The starting Pd nanoparticles with a size of 1.8 ± 0.3 nm are embedded within the titania layers. X-ray diffraction, X-ray photoemission spectroscopy and TEM characterization of the final hybrid materials[26], and comparison with that of the **Pd/TiO₂** and of the starting protected Pd nanoparticles confirm the formation of homogeneous hierarchical materials[30]. It is important to note that for both **Ox-MWCNTs@Pd/TiO₂** and **Tour-MWCNTs@Pd/TiO₂**, the coverage with the titania layer is not complete, but zones of bare MWCNTs are clearly observed (Fig. 2d). This is an advantage as it provides the hybrid material with points of direct contact between the nanotubes and the electrode, securing an improved electric conductivity within the nanocomposite and an efficient electron transfer to the inorganic phase.

In the as-prepared hybrids **Ox-MWCNTs@Pd/TiO₂** and **Tour-MWCNTs@Pd/TiO₂**, the titania layer is in its amorphous phase, as we failed to find any TiO₂ crystal grains by

high-resolution TEM (HRTEM). Raman analyses of the sample confirmed that the titania layer is not crystalline, as no peaks for distinctive crystal phases were found in any area. The integration of the amorphous TiO₂ was however confirmed by Fourier transform infrared, where together with the signature C=C stretching at 1,625 cm⁻¹ of the CNTs, we observed appearance of the characteristic Ti–O–Ti vibration modes at 700 and 672 cm⁻¹ after reaction with the inorganic **Pd/TiO₂** (Supplementary Figs 3 and 4). The amorphous state of the TiO₂ may have important implications in enhancing the affinity for water molecules, thanks to the presence of Ti–OH groups. On the other hand, as revealed by HRTEM, the thermal annealing treatment causes a transition to the anatase phase with crystallites size that averages 10 nm (Fig. 2e; Supplementary Fig. 5). The annealing also removes all the organic groups from the material, tightening the contact between the MWCNTs and the TiO₂, which is crucial to enhance the catalytic performance as a result of the increased level of heterojunctions (Supplementary Fig. 6). Because in the calcined samples the organic spacers are now removed, we expect that the structural differences between the Ox-MWCNTs and Tour-MWCNTs nanscaffolds are no longer present, so that they are likely to converge towards an analogous HER material as it has been confirmed by the electrocatalytic studies (see discussion below).

From the tomographic reconstruction (Fig. 1b), the Pd nanoparticles are also evident, as resulting from the higher contrast in the scanning transmission electron microscopy (STEM) projections due to the high atomic number of Pd with respect to the other elements. However, due to the experimental missing wedge in sample rotation during acquisition (sample tilt −60°/+60°), they appear slightly elongated in the direction perpendicular to the wire axis in the reconstruction.

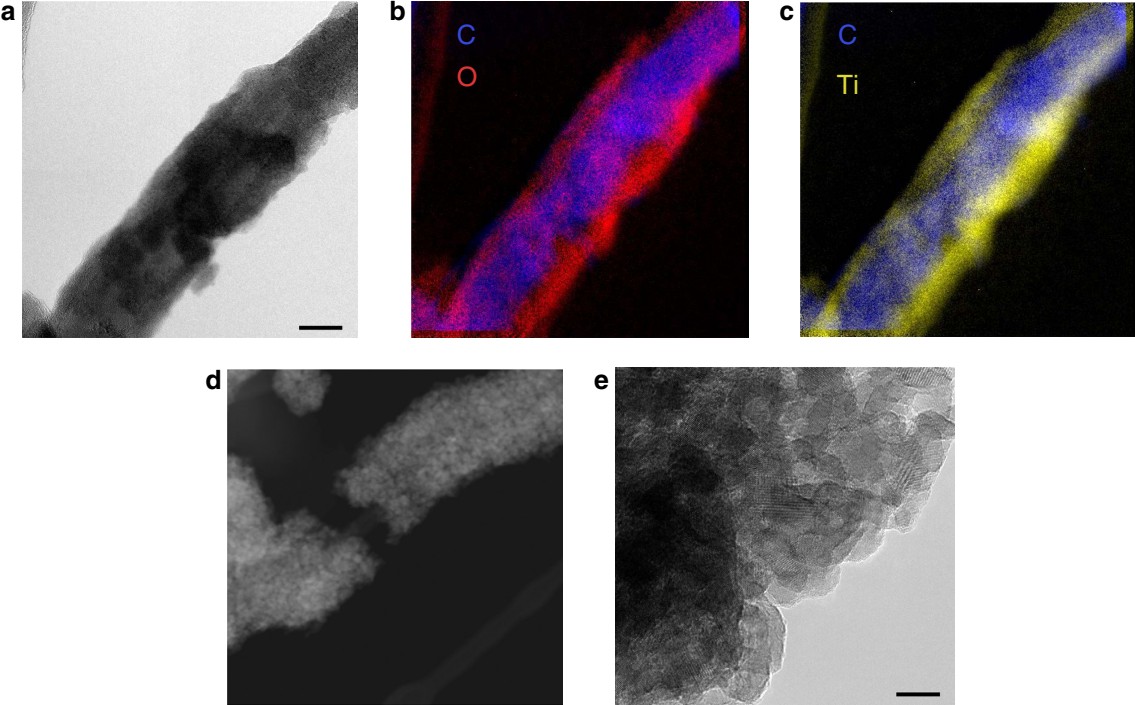

**Figure 2 | Characterization of MWCNTs@Pd/TiO₂ nanohybrids.** (**a**) Representative TEM image of the **f-MWCNTs@Pd/TiO₂** nanoarchitectures prepared by self-assembly, where f-MWCNTs are embedded inside a layer of TiO₂ that contains Pd nanoparticles dispersed within. Scale bar, 20 nm. (**b,c**) Energy filtered TEM images highlighting the elemental distribution of the TiO₂ shell (red and yellow for O and Ti atoms, respectively) surrounding the f-MWCNT backbone (C atoms in blue). (**d**) STEM-HAADF of a typical **Ox-MWCNT@Pd/TiO₂**. (**e**) Large magnification HRTEM of an area of a thermally annealed **Ox-MWCNT@Pd/TiO₂** showing the formation of crystalline TiO₂ anatase phase (scale bar, 10 nm).

**Electrochemical properties.** The electrochemical investigation of the annealed nanohybrids, **f-MWCNTs@Pd/TiO₂**, evidenced remarkable changes in the electronic structure of TiO₂ caused by its integration with MWCNTs, changes that will in turn favourably contribute to the enhancement of the electrocatalytic properties of the hierarchical **Pd/TiO₂** structure. An increase of (i) the number of free electrons in the titania conduction band and (ii) surface defect density were in fact highlighted by the analysis of the interfacial capacity ($C_{sc}$) of **f-MWCNTs@Pd/TiO₂** compared with pristine **Pd/TiO₂**. In Fig. 3a, the $C_{sc}$ values, measured by electrochemical impedance spectroscopy, are plotted according to the classical Mott–Schottky (M–S) equation[31]:

$$\frac{1}{C_{sc}^2} = \frac{2}{\varepsilon\varepsilon_0 N_D}\left(E - E_{fb} - \frac{k_B T}{e}\right),\qquad(1)$$

where $\varepsilon$ is the relative dielectric constant, $\varepsilon_0$ the permittivity of vacuum, $e$ the electronic charge, $k_B$ the Boltzmann constant. From the analysis of the narrow $E$-dependent linear region ($E \sim 0.00$ V), a flat-band potential $E_{fb} = -0.22$ V was obtained for all systems, in good agreement with the reported values for TiO₂ at neutral pH[32]. However, on passing from **Pd/TiO₂** to **f-MWCNT@Pd/TiO₂**, the apparent doping density ($N_D$), and hence the amount of free electrons in the TiO₂ conduction band, increased by ∼600 times and it further depends on the f-MWCNT concentration (Supplementary Figs 7 and 8; Supplementary Table 1)[33]. The larger availability of free electrons at the MWCNT/TiO₂ interface is expected to play a favourable effect on the HER kinetics, especially at low overpotentials.

Finally, the analysis of the M–S plots at more positive potentials, where the capacity attains a constant value, provided evidence for the presence of extended surface states, whose density (proportional to $C_{sc}$) largely increased on passing from **Pd/TiO₂** to **f-MWCNTs@Pd/TiO₂** (Supplementary Table 1)[34].

Such defects on the surface of the titania nanocrystals (oxygen vacancies and incompletely coordinated Ti atoms)[35], that would be mostly localized at the interface with MWCNTs, may provide efficient catalytic sites for the adsorption and catalytic dissociation of water molecules, an important step for the promotion of HER in neutral and alkaline media (H_{ad} formation). As a matter of fact, as the kinetic studies of **f-MWCNTs@Pd/TiO₂** show (Tafel slope analysis and [H⁺] dependence; Supplementary Figs 9–11), water dissociation and adsorbed hydrogen formation would represent the rate-determining step of the HER in the present conditions.

**Electrocatalytic performance of HER.** The HER activity of the nanohybrids was investigated by linear sweep voltammetry and chronoamperometry (CA), in de-aerated phosphate buffer solutions at pH 7.4 (Fig. 3b,c), using a standard three-electrode electrochemical cell, equipped with an Ag/AgCl (3 M) reference electrode and a platinum counter electrode. The hydrogen-evolving activity of the catalyst was assessed qualitatively and quantitatively using a scanning electrochemical microscope (SECM)[36,37]. The molecular hydrogen probe, namely, a platinum ultramicrometric tip biased at 0.4 V (versus RHE), was located at few micrometres from the electrode surface, under conditions of constant hydrogen collection efficiency (Supplementary Fig. 12). The SECM screening method was used to probe a few mm² surface providing a faster and more direct way to quantify the electrocatalytic efficiency of the reactive material with respect to standard (colorimetric and chromatographic) methods.

The catalytic onset for the HER of both **Ox-MWCNTs@Pd/TiO₂** and **Tour-MWCNTs@Pd/TiO₂** electrodes is located at $E < -0.2$ V, while the overpotentials measured at an arbitrary cathodic current threshold of 1 mA cm⁻² (corresponding to a hydrogen evolution rate of 19 μmol h⁻¹ cm⁻²) were −0.31 and

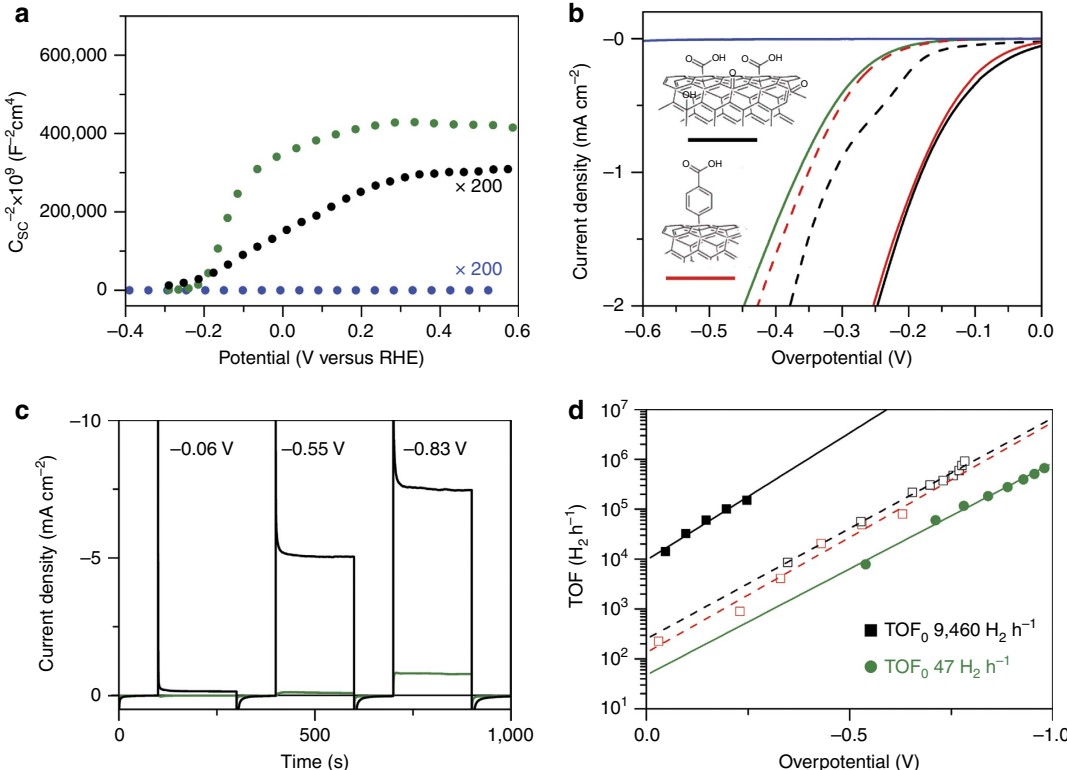

**Figure 3 | Electrocatalysis of HER on nanohybrid cathodes.** Films were prepared by drop-casting a controlled amount of the catalyst inks onto either carbon screen-printed electrode (SPE) or glassy carbon (GC) electrodes (both substrates are catalytically inert towards the HER process). (**a**) Mott–Schottky plots ($C^{-2}$ versus potential) for the various nanostructured systems in phosphate buffer, pH = 7.4, obtained at a fixed frequency of 1 kHz: **f-MWCNT@Pd/TiO$_2$** (black circle), **Pd/TiO$_2$** (green circle) and the pristine electrode surface (blue circle). (**b**) Linear sweep voltammetry and (**c**) chronoamperometric curves in phosphate buffer (pH 7.4) for the pristine SPE electrode (blue line), after the deposition of the nanocomposites **Ox-MWCNT@Pd/TiO$_2$** (dashed black line), **Tour-MWCNT@Pd/TiO$_2$** (dashed red line), **Pd/TiO$_2$** (green line) and calcined **f-MWCNT@Pd/TiO$_2$** (solid black and red lines) Scan rate = 2 mV s$^{-1}$. (**d**) Electrocatalytic performances for **Ox-MWCNT@Pd/TiO$_2$** (dashed black line), **Tour-MWCNT@Pd/TiO$_2$** (dashed red line), **Pd/TiO$_2$** (green line) and calcined **f-MWCNT@Pd/TiO$_2$** (solid black line): TOF as a function of $\eta$. $\eta = E_{app} - E^0 - i_{ss} \times R_s$, where $E_{app}$ is the applied potential, $E^0$ is the thermodynamic potential corrected for the pH ($E^0$ [H$^+$/H$_2$] = −0.21–0.059 × pH = −0.65 V versus Ag/AgCl and $i_{ss} \times R_s$ is the correction for the ohmic drop ($i_{ss}$ is the stationary state current measured during the potential step and $R_S$ is the uncompensated resistance). TOF$_0$ is the turnover frequency at zero overpotential (TOF at $\eta$ = 0 V). All potentials are reported versus RHE.

−0.35 V, respectively (Fig. 3b). We attribute such differences in the overpotentials to the organic spacer between the two nanocarbons and the titania. Namely, the benzoic groups separate the titania layer from the nanotubes more than simple carboxylic or hydroxyl groups, leading to a less efficient electronic communication. The optimized interface was obtained by removing the organic functionalization with a thermic treatment. As previously mentioned, calcination at 350 °C for 3 h increase the interaction between the MWCNT and TiO$_2$, resulting in significantly higher HER current and higher apparent doping density (see Fig. 3b the comparison between dashed and solid lines; Supplementary Fig. 8; Supplementary Table 1). After calcination, **Ox-MWCNTs@Pd/TiO$_2$** and **Tour-MWCNTs@Pd/TiO$_2$** merge to a structure with the same catalytic property: **f-MWCNTs@Pd/TiO$_2$**. A general remarkable improvement in the HER performance was observed, as evidenced by the positive shift of $\eta$ (at 1 mA cm$^{-2}$) to a value of −0.17 V (*vis-à-vis* to −0.37 V in the case of **Pd/TiO$_2$**). Notice that, under the above conditions, protons are reduced on massive platinum electrodes at about −0.10 V and that at this pH about 50% of the overpotential is due to the concentration overpotential[38,39]. The Tafel slopes of ~130 mV per decade (Supplementary Fig. 9; Supplementary Table 1) were in good agreement with literature values for HER at Pd electrodes[40], and, combined with the unitary slope observed in the log(i)/log[H$^+$] plot (experiments in non-aqueous media,

Supplementary Figs 10 and 11), emphasize the role of the Volmer step (water dissociation and formation of reactive intermediate H$_{ad}$: H$_2$O + Pd + e = Pd − H$_{ad}$ + OH$^-$) as the rate-determining step[41]. The availability of free protons in neutral and alkaline media is critical and, in such a scenario, the titania shells would promote the adsorption of water molecule and its dissociative adsorption at the boundaries with the Pd clusters, as depicted schematically in Fig. 1c. A similar mechanism was recently reported in bifunctional Pt-Ni(OH)$_2$ nanosystems to explain their efficient promotion of electrocatalytic HER in alkaline media[8]. It can also be inferred that the observed increase of titania surface defects, associated to MWCNTs, would enhance the dissociative adsorption process, thus explaining the increased performance of **f-MWCNTs@Pd/TiO$_2$** nanohybrid systems.

**TOF quantification.** Quantification of electrocatalytic activity was performed by the combination of different electrochemical techniques. The current transients measured in the CA experiments (Fig. 3c), in which the cathode potential was stepped back and forth between 0 V and a negative value in the range −0.5/−1.0 V, were stable and reproducible. Importantly, all catalysts displayed great chemical and mechanical stability (even under massive hydrogen evolution conditions) without any significant activity loss after repetitive cycling and after >45 h

operation (<15% activity loss; Supplementary Fig. 13). The TOF at each overpotential $\eta$, that is, the moles of hydrogen generated by the electroactive Pd catalytic sites in the time unit, was then derived as described in equation (2) from the integrated currents at the various substrates ($Q_{H_2}^{\eta,\mathbf{MWCNT@Pd/TiO_2}}$) after subtraction of the background contribution ($Q_{H_2}^{\eta,\mathbf{MWCNT@TiO_2}}$, measured in the absence of Pd nanoparticles) and after normalization by the amount of electroactive Pd ($Q_{Pd}$; Supplementary Fig. 14).

$$\mathrm{TOF}_{\eta} = \frac{\left(Q_{H_2}^{\eta,\mathbf{MWCNT@Pd/TiO_2}} - Q_{H_2}^{\eta,\mathbf{MWCNT@TiO_2}}\right)}{Q_{Pd} \times \mathrm{time}} \times \mathrm{FE} \quad (2)$$

The faradaic efficiency (FE) term in equation (2), that is, the fraction of cathodic current that actually generates hydrogen, was determined *in situ* during electrolysis using the SECM-Pt hydrogen probe (Supplementary Fig. 15). A preliminary determination of the collection efficiency of the probe was obtained following a reported procedure[42], and thoroughly described in Supplementary Notes 2 and 4 (Supplementary Figs 16 and 17). In Fig. 3d, the TOF values for the various nanohybrids are finally reported considering a FE of 90% (Supplementary Notes 3 and 4): the Tafel-like behaviour demonstrates the good stability of the catalytic film and a well-behaved ohmic contact to the underlying conducting substrate. The electrocatalytic performance of the optimized **f-MWCNTs@Pd/TiO_2** systems resulted in an outstanding TOF of 15000 $H_2$ per hour at $\eta = 50$ mV.

## Discussion

Hydrogen is considered one of the viable alternatives to fossil fuels. However, its sustainable production from water splitting is still hampered by major thermodynamic and kinetic hurdles of electrocatalysis and energy materials. In fact, despite the huge research efforts in this area, $H_2$ production by water electrolysis occurs with prohibitive overpotentials and requires >1.8 V applied voltage, together with harsh strong alkaline or acidic pH conditions. The unsolved problem of the HER calls for the engineering of Pt-free catalyst that can work with low overpotential and remarkable activity, in unfavourable conditions of $H^+$ concentration, that is, at neutral pH. In this scenario, our results provide an important step forward in the design of electrocatalytic nanomaterials for HER. A direct comparison with state-of-the-art heterogeneous HER electrocatalysts that operate at neutral pH (Fig. 4) ranks the thermal annealed nanohybrids **f-MWCNTs@Pd/TiO_2** among the best performing materials so far reported (see Supplementary Table 2 for an extensive comparison of TOF, exchange current density and Tafel slope values) and demonstrates the ability to operate at a very high rate at low overpotential.

Furthermore, extrapolation of the TOF-$\eta$ semilogaritmic plot to zero overpotential provided the $TOF_0$ intercept value, which is an optimal descriptor of the intrinsic performance of electrocatalysts[43]. For **f-MWCNTs@Pd/TiO_2**, the $TOF_0$ was 9,460 $H_2$ per hour, whereas in the case of **Pd/TiO_2**, a significantly lower $TOF_0$ value of 47 $H_2$ per hour was measured, thus confirming the important role played by the carbon nanomaterials in promoting the catalytic activity of the metal nanoparticles. By changing the doping level and surface state density of the titania layers through hybridization with MWCNTs, a marked enhancement of the electrocatalytic properties of the nanocomposites is observed, as it is evident from the comparison between catalysts with different MWCNTs concentrations (Supplementary Fig. 18). In this respect, films obtained by physically mixing **Ox-MWCNT** with **Pd/TiO_2** and **Ox-MWCNT** with **Pd/TiO_2** or the composite **f-MWCNTs@Pd** (Supplementary Fig. 19) gave significantly higher capacitive currents but much worse electrocatalytic activities and lower stability (Supplementary Figs 20 and 21).

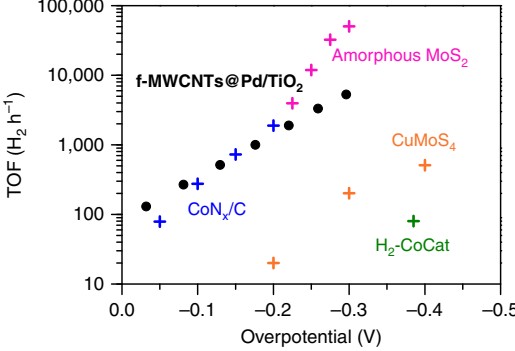

**Figure 4 | Comparison of mass-normalized TOF with other catalysts.** Direct comparison of TOF plots for **f-MWCNTs@Pd/TiO_2** nanohybrid together with other recently reported heterogeneous HER electrocatalysts that operate at neutral pH. The TOF herein reported are calculated considering the entire catalyst loading on the various electrodes and using rotating disk glassy carbon electrode (rotation speed of 1,600 r.p.m.). The data are adapted from: ref. 47 for $CoN_x/C$ (2015), ref. 48 for amorphous $MoS_2$ (2011), ref. 49 for $H_2$-CoCat (2012) and ref. 50 for $CuMoS_4$ (2012).

This highlights the synergistic effects and the advantages derived from the nanoscale integration of the various components in the **f-MWCNTs@Pd/TiO_2** nanohybrids. Moreover, the 50 times higher $TOF_0$ values of **f-MWCNTs@Pd/TiO_2** with respect to **Ox-MWCNTs@Pd/TiO_2** and **Tour-MWCNTs@Pd/TiO_2** highlight again the essential role of the MWCNT/TiO_2 interface to promote an efficient HER process.

As schematically depicted in Fig. 5, MWCNTs would therefore promote the electrocatalytic activity of the **Pd/TiO_2** by two concurrent electronic effects: (i) since the flat-band potential of TiO_2 is only 220 mV more negative than the potential of the hydrogen electrode[32], the amount of free electrons at the titania surface that may promote the HER at low overpotentials is rather small. In such conditions, the increased doping level induced by MWCNTs (Supplementary Table 1) is going to exert a strongly positive effect on the HER kinetics. In addition, (ii) the presence of extensive surface states at the TiO_2/MWCNTs interface would promote a better coupling with the Pd electronic levels, globally resulting in faster kinetics of the HER process.

In particular, we have designed a three-component interface, where MWCNT template the one-dimensional morphology of a **Pd/TiO_2** units active layer. The catalyst composition and morphology are shaped to leverage the key steps of the HER mechanism in neutral pH conditions, namely: (i) $H_2O$ binding and electron-induced dissociation at the **MWCNT@TiO_2** metal oxide surface; (ii) stabilization of the atomic hydrogen intermediate on Pd; and (iii) hydrogen evolution by reaction of the adsorbed atomic hydrogen at the **Pd/TiO_2** interface. This functional synergy stems from a favourable interplay of kinetic and thermodynamic factors, yielding an efficient Pt-free catalyst package with (i) notable performances in neutral water media ($TOF_0$ of 9,460 $H_2$ per hour, negligible onset potential for HER, exchange current density of $6 \times 10^{-5}$ A cm$^{-2}$), (ii) excellent robustness (current decay <15% after 45 h electrolysis) and (iii) strong correlation between HER activity and the electronic characteristics of the MWCNT/TiO_2 interface (50 times faster HER after optimization via thermal annealing).

To summarize, we have presented evidence on how the hierarchical integration of multi-walled CNTs and metal/metal oxides within a single nanostructure can leverage the potentiality of HER electrocatalysis at neutral pH. Moreover, the modularity of the catalyst fabrication is open to a straightforward fine-tuning of the metal–metal oxide pairs and nanoscaffolds to meet

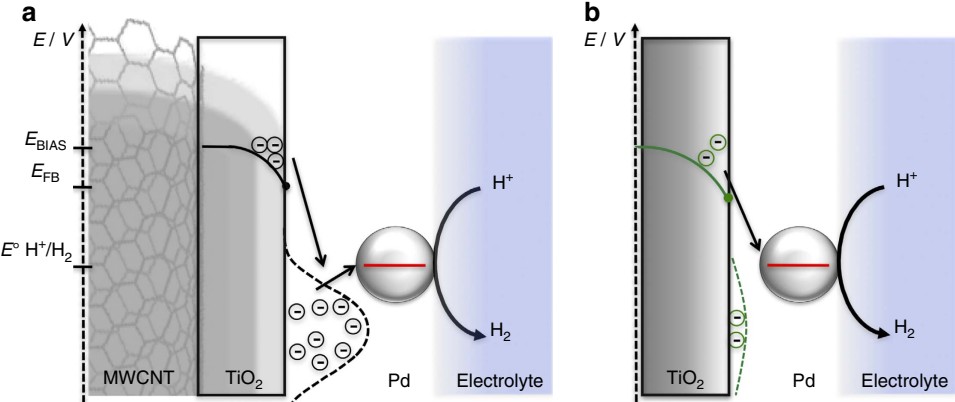

**Figure 5 | Schematic energy level representation.** Band-bending diagram and schematic representation of the energy levels involved in the HER process. The scheme also depicts qualitatively the surface states distributions: (**a**) **f-MWCNTs@Pd/TiO$_2$** and (**b**) **Pd/TiO$_2$** (green dashed line). $E_{BIAS}$: $-0.35$ V; $E_{fb}$: $-0.22$ V; $E°$ [H$^+$/H$_2$]: 0 V (versus RHE).

the HER requirements spanning a wide range of experimental conditions. This approach is expected to provide insights in the HER electrocatalytic mechanisms while implementing versatility and flexibility of the catalyst package for a sustainable hydrogen economy.

## Methods

**Chemicals.** All glassware was dried in an oven set to a temperature of 80 °C for 24 h before use. All reagents were purchased from Sigma Aldrich and used without further purification. Phosphate buffer (PB) electrolytes were prepared readily before experiments from reagent-grade chemicals, mixing appropriate amount of NaH$_2$PO$_4$ and Na$_2$HPO$_4$ salts.

**Electrochemistry.** Linear sweep voltammetry, cyclic voltammetry (CV), chron-oamperometry (CA) and electrochemical impedance spectroscopy experiments were carried out in a three-electrode electrochemical cell using as working electrodes screen-printed electrodes (supplied by DROPSENS S.L., model DS110 diameter 4 mm), glassy carbon electrodes (CH Instruments, diameter 3 mm) or on indium tin oxide (purchased from Kuramoto Seisakusho Co., Ltd, Tokyo, Japan) as a working electrode. A Biologic SP300 potentiostat was used as the work station for all the electrochemical experiments. Thin films of the different nanocomposites were prepared on the substrate working electrodes by deposition of the respective catalyst inks (methanol suspensions, 1.6 mg ml$^{-1}$). The best performing films (Supplementary Fig. 22) combine good conductivity and an optimum catalyst loading (170 µg cm$^{-2}$). Thin compact films were obtained with 5 µl aliquots subsequently added until a total deposited volume of 100 µl was reached. Those were then allowed to dry in the dark for at least 12 h and the film thickness on the electrode surface, measured with a Tencor AlphaStep profilometer, was 10 ± 3 µm with a roughness of 3 ± 1 µm.

**Transmission electron microscopy.** HRTEM and STEM in high-angle annular dark-field mode (HAADF) were performed on a JEOL JEM-2200FS microscope, working at 200 kV. Energy filtered maps were obtaining using an in-column energy spectrometer (Ω-type). In the STEM-HAADF images (as for the tomography projections), the camera length value was set to assure an inner cutoff angle of the STEM detector >100 mrad, to reduce the contribution from coherent contrast to the minimum and to enhance the contrast of Pd particles with respect to TiO$_2$ (higher atomic number of Pd with respect to Ti). The three-dimensional tomography was obtained by a simultaneous iterative reconstruction technique (SIRT) reconstruction (20 iterations) of a series of 46 projections acquired in STEM-HAADF (from $-30$ to $+60°$ with 2° step).

**Experiments in non-aqueous media.** Electrochemical experiments were carried out in an airtight single-compartment cell described elsewhere, using platinum and silver spirals, respectively, as counter and quasi-reference electrodes[44]. The $E_{1/2}$ values, reported in the Ag/AgCl/KCl (3 M) scale, have been determined by adding decamethylferrocene as an internal standard at the end of each experiment. All the experiments were conducted in anhydrous acetonitrile and drying the supporting electrolyte, tetrabutylammonium hexafluorophosphate TBAPF$_6$, for 60 h under vacuum before each experiments. Additions of [DMFH](OTf):DMF (1 mol:1mol) were made from a solution with a [DMFH]$^+$ concentration of 4.1 M, following a procedure previously described[24]. All the solutions were argon-saturated for 30 min before performing the experiments.

**Experiments in aqueous media.** Electrochemical experiments were carried out in a home-made cell having a 5 mm diameter opening in front of the substrate electrodes. An O-ring ensures a perfect tightening of the assembly thanks to connecting screws that fit directly into the body of the cell. All the experiments were carried out in phosphate buffer 0.1 M (pH = 7.4) freshly prepared readily before each experiment. The electrochemical cell was equipped with a platinum spiral counter electrode and a Ag/AgCl/KCl (3 M) reference electrode. Potentials were then reported in the reversible hydrogen electrode scale by shifting the potentials by ($E_{RHE} = E_{Ag/AgCl} + 0.436$ V). All the solutions were thoroughly degassed with argon for at least 25 min before each experiment. Comparison of TOF performances for **f-MWCNTs@Pd/TiO$_2$** and other state-of-the-art HER electrocatalysts was done in a rotating disk electrode (glassy carbon) configuration (Fig. 4). CA measurements have been performed at a rotation speed of 1,600 r.p.m. and under constant Ar flow. For a straightforward comparison of the catalysts, mass-normalized TOF data were obtained considering the entire loading of Pd on the electrode (1 wt.%) and not on an electroactive area basis.

In the M–S analysis, we scanned the frequency from 1 kHz to 1 Hz (three frequencies per decade) and we varied the potential in the range from 0.8 to $-0.4$ V (versus RHE). All the experiments were conducted in a phosphate buffer solution at pH = 7.4. The following constants were used for the M–S data processing: $k_B = 1.38 \times 10^{-23}$ J K$^{-1}$, $T = 298$ K, $e = 1.6 \times 10^{-19}$ C, $\varepsilon_0 = 8.86 \times 10^{-12}$ F m$^{-1}$, $\varepsilon = 48$ F m$^{-1}$ for anatase TiO$_2$ (ref. 45).

For the SECM experiments, a CH Instruments CHI 910B SECM apparatus was used. A 15 µm platinum ultramicroelectrode (CHI Instruments, USA) was used as the tip working electrode. The approaching curves to the substrates were obtained via feedback mode in a classic three-electrode cell, using 1 mM ferrocenemethanol in a 100 mM LiClO$_4$ aqueous solution. The surface generation-tip collection experiments were carried out in argon-saturated solution (pH 7.4)[46].

**Data availability.** The data that support the findings of this study are available from the corresponding authors on request.

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

## Acknowledgements

This work is supported by the Italian Ministero dell'Istruzione, Università e Ricerca (FIRB RBAP11C58Y, PRIN-2010N3T9M4, PRIN-2009Z9ASCA and FIRB RBAP11-ETKA_006), Universities of Bologna and Trieste, INSTM, the Seventh Framework Programme [FP7/2007–2013] under grant agreement no. 310651 (SACS project), EU Cost action CM1104.

## Author contributions

G.V. and A.B. contributed equally to this work. M.Me. and M.C. performed the synthetic tasks; L.N. and G.B. performed the HRTEM and STEM analysis; G.V. and A.B. optimized the deposition protocols, performed and analysed the electrochemical and electrocatalytic experiments; G.V., A.B. and M.Ma., performed the electrochemistry investigation in non-aqueous media; G.V., A.B. and S.R. performed the scanning electrochemical experiments; R.J.G., M.B., M.P., P.F. and F.P. planned and supervised the research and co-wrote the paper with contributions from other authors. All authors discussed the results and commented the manuscript.

## Additional information

**Competing financial interests:** The authors declare no competing financial interests.

