## [Peer Review File · Nature Communications]

Reviewers' comments:

Reviewer #1 (Remarks to the Author):

Increasing the efficiency of H₂ production using water as the primary source by various efforts has become a hot topic in recent years in the world. In this manuscript, the authors have constructed a hierarchical, co-axial arrangement of a Pd/ TiO₂ outer layer on functionalized multi-walled carbon nanotubes (f-MWCNT). Such 3D-electrocatalytic electrode is able to merge a highly conductive nano-carbon core with the active Pd/TiO₂ hetero-phase, to promote the water dissociation, hydride formation and H₂ evolution. And they claimed that the as-prepared f-MWCNTs@Pd/TiO₂ exhibited a superior electrocatalytic performance, with turnover frequency TOF of 150.000 H₂ h⁻¹ at an applied overpotential as low as 50 mV, far exceeding the activity of state-of-the-art HER electrocatalysts. The f-MWCNTs@Pd/TiO₂ is a novel structure of electrocatalyst based on the functionalized multi-walled carbon nanotubes, though the nano Pd/TiO₂ had been already developed as the cathode for H₂ production. It is attractive to see that the efficiency of H₂ production by such hierarchical electrocatalyst reached a top level comparing to the previously reported HER electrocatalysts. The authors have well explained the synergic mechanism of HER by f-MWCNTs@Pd/TiO₂ electrocatalytic cathode based on the water dissociation, formation of the hydride intermediate and H₂ evolution. So I would like to recommend to be accepted by the journal, however some questions most about experiment in the manuscript should be addressed.

- (1) How thick is the compact film of f-MWCNTs@Pd/TiO₂ on the substrate of working electrodes? Does it have any effect on the HER behavior?
- (2) How is the bonding strength of f-MWCNTs@Pd/TiO₂ film on the substrates during the H₂ evolution? What difference of HER behavior for three different kind of working electrodes?
- (3) The authors are suggested to provide HRTEM images for core/shell structure of Pd/TiO₂, from Fig.2(a) it is not clear to identify the Pd/TiO₂ core/shell structure.
- (4) Can the authors make any experimental comparison of HER properties for the amorphous and crystalline TiO₂?

Reviewer #2 (Remarks to the Author):

In this paper, Giovanni Valenti et al. reported the hierarchical, co-axial arrangement of a Pd/ TiO₂ outer layer on functionalized multi-walled carbon nanotubes (f-MWCNT) and their study for catalytic H₂ evolution. Despite the detailed examination of this system I do not see much significant novelties in their work. First, using carbon nanotubes (MWNT or even SWNT) as electronic mediator for HER reaction have been widely reported in the literature (including the previous works from some of the present authors). The importance of forming effective interfaces with noble metals/oxide or MoS₂ (as Pt substituent) have also been well noted. The methodologies and chemistry for immobilization onto MWNT used in this paper have been widely studied. The authors commented on the replacement of Pt with Pd for this potential application. Although the availability of Pd is slightly higher than Pt, the benefits of replacing Pt with Pd are not great (of a related market and Pd is sometimes more costly than Pt) compared with other non-Pt substituents. I do not think it will make much improvement in this situation. The authors also reported a more superior performance when the immobilised Pd/TiO₂ on MWNT was subjected to calcination/oxidation than those without. What would happen to the organic interface upon calcination? would it be severely reconstructed to lead to aggregation? There was no metal and TiO₂ particle size analysis before and after the heat treatment or any detailed characterization of the interface. The conventional metal surface area measurements by chemisorption or ECSA by electrochemical techniques may be useful to be included. I have unfortunately found the resolution of the presented images rather low and not informative. Overall, I do not recommend this paper to be published in Nature Commun. due to the lack of novelty. I suggest this work should be submitted to more specialised electrochemical or energy journals.

Reviewer #3 (Remarks to the Author):

In this work, the authors have demonstrated an assembly of a 3D-electrocatalytic interface, featuring a hierarchical, co-axial arrangement of a Pd/TiO₂ outer layer on multi-walled carbon nanotubes. The resultant Pd/TiO₂/MWCNT exhibited enhanced HER performance compared with Pd/TiO₂. Overall, it is a nice piece of work and can be considered for publication in Nature Communications after addressing the following points:

1. In the introduction, authors demonstrate that Pt-based catalysts are plagued by the unfavorable market price. However, Pd is also as expensive as Pt.
2. f-MWCNTs@Pd/TiO₂ reaches a turnover frequency at zero overpotential, TOF₀, of 9460 H₂ h⁻¹. At zero overpotential, the HER reaction does not take place, why the TOF is so high?
3. What is a co-axial TiO₂ outer layer?
4. In Figure 2, where are the Pd nanoparticles? Could the authors provide the HR-TEM and STEM mapping images? How to confirm the size of Pd nanoparticles?
5. "It is important to note that for both Ox-MWCNTs@Pd/TiO₂ and Tour-MWCNTs@Pd/TiO₂ the coverage with the titania layer is not complete, but zones of bare MWCNTs are clearly observed (Fig. 2d)." Again, the authors shall provide HR-TEM images, where the interfaces between TiO₂ and Pd nanoparticles can be directly identified?
6. What is the active site if the Pd nanoparticles are embedded within the titania layers?
7. "Such defects on the surface of the titania nanocrystals (oxygen vacancies and incompletely coordinated Ti atoms),..... water dissociation and adsorbed hydrogen formation would represent the rate determining step of the HER in the present conditions." Where are the experimental and computational proofs?
8. In SI, the HER mechanism seems not be correct and the first step should be the water dissociation (Volmer step).
9. The enhance HER activity of f-MWCNT@Pd/TiO₂ should originate from the improved conductivity after combining MWCNT. The authors may clarify this point.
10. The HER activity of Pd/MWCNT should be provided.

Referee # 1

Increasing the efficiency of H₂ production using water as the primary source by various efforts has become a hot topic in recent years in the world. In this manuscript, the authors have constructed a hierarchical, co-axial arrangement of a Pd/ TiO₂ outer layer on functionalized multi-walled carbon nanotubes (f-MWCNT). Such 3D-electrocatalytic electrode is able to merge a highly conductive nano-carbon core with the active Pd/TiO₂ hetero-phase, to promote the water dissociation, hydride formation and H₂ evolution. And they claimed that the as-prepared f-MWCNTs@Pd/TiO₂ exhibited a superior electrocatalytic performance, with turnover frequency TOF of 150.000 H₂ h⁻¹ at an applied overpotential as low as 50 mV, far exceeding the activity of state-of-the-art HER electrocatalysts. The f-MWCNTs@Pd/TiO₂ is a novel structure of electrocatalyst based on the functionalized multi-walled carbon nanotubes, though the nano Pd/TiO₂ had been already developed as the cathode for H₂ production. It is attractive to see that the efficiency of H₂ production by such hierarchical electrocatalyst reached a top level comparing to the previously reported HER electrocatalysts. The authors have well explained the synergic mechanism of HER by f-MWCNTs@Pd/TiO₂ electrocatalytic cathode based on the water dissociation, formation of the hydride intermediate and H₂ evolution. So I would like to recommend to be accepted by the journal, however some questions most about experiment in the manuscript should be addressed.

1) *“How thick is the compact film of f-MWCNTs@Pd/TiO₂ on the substrate of working electrodes? Does it have any effect on the HER behavior?”*

The thickness of the electro-catalytic film was optimized by considering the impact of the catalyst loading on the electrode performance. To better clarify this aspect, Figure S7a in SI has been added in the revised version of the manuscript. The figure shows the linear sweep voltammetry curves (phosphate buffer, pH 7.4) obtained using different amount of f-MWCNTs@Pd/TiO₂, together with the analysis of the film resistance by impedance spectroscopy experiments. Peak HER performance stems from the best compromise between the film deposition (catalyst loading of 170 μg cm⁻²) and the resulting conductivity properties

of the functional electrode. The optimized deposition protocol leads to a mean film thickness of $10 \pm 3 \mu\text{m}$ measured with a Tencor AlphaStep profilometer, and a surface roughness of $3 \pm 1 \mu\text{m}$. This evidence is now reported in the main text (Page 14) and in SI (Paragraph 4).

2) *"How is the bonding strength of f-MWCNTs@Pd/TiO₂ film on the substrates during the H₂ evolution?"*

The chemical and mechanical integrity of the f-MWCNTs@Pd/TiO₂ electrocatalyst, has been tested for more than 45 h, under HER conditions. The catalytic current decreases less than 15% under these conditions (Figure S18). This observation is now highlighted in the main text (page 10) and in SI (page 38).

3) *"What difference of HER behavior for three different kind of working electrodes?"*

The three functional electrodes, namely Ox- (or Tour-) MWCNT@Pd/TiO₂, Pd/TiO₂, and f-MWCNTs@Pd/TiO₂, have been compared considering: (i) the Tafel analysis; (ii) the resulting J_0 , TOF_0 , and onset overpotential parameters; (iii) the long-term operation stability. Our results now include the Tafel-slope determination for all the electrocatalytic materials under investigation. In all cases, a similar Tafel-slope value of $\sim 130 \text{ mV/decade}$ (new Figure S10 and Table S1) are consistent with a mechanism involving the Pd-mediated water dissociation (Volmer step $\text{H}_{\text{ad}}: \text{H}_2\text{O} + \text{Pd} + \text{e}^- = \text{Pd-H}_{\text{ad}} + \text{OH}^-$) as the rate-determining step.

The superior performance of the three-component interface has been thoroughly addressed in the manuscript, in terms of multiple electronic effects (surface defects, doping density, etc...). As a general remark, the HER performances of the different nanohybrids, has been obtained in repeated experiments, showing reproducible LSV behaviour, as shown in the new Figure S7b (representative comparison of dashed and solid black line).

4) *"The authors are suggested to provide HRTEM images for core/shell structure of Pd/TiO₂, from Fig.2(a) it is not clear to identify the Pd/TiO₂ core/shell structure."*

The referee correctly indicates that proving of the core-shell nature of Pd@TiO₂ is not a trivial task. The concept of self-assembly of protected Pd nanoparticles within a surrounding metal oxide shell composed of CeO₂, TiO₂ or ZrO₂ nanocrystals was previously postulated for analogous synthesis ([1] J. Am. Chem. Soc. 2010, 132, 1402, [2] Science 2012, 337, 713, [3] ChemSusChem 2012, 5, 140). Very recently, similar systems have shown to be more complex, presenting some heterogeneity and being subjected to modification upon thermal treatments

(Nature Communications 2015, 6, art. 7778). In line with the reviewer's concern, we modified the manuscript by replacing the concept of core-shell and discussing the electrocatalyst structure in terms of the tight metal-metal oxide contact obtained by hierarchical self-assembly synthesized from protected Pd nanoparticles, Ti alcoxide, and functionalized nanotubes.

5) *"Can the authors make any experimental comparison of HER properties for the amorphous and crystalline TiO₂?"*

We thank the reviewer for raising this important issue. Bare amorphous and crystalline TiO₂ are both poorly active in HER. On the other hand, the performances of the amorphous (before calcination) and crystalline (after calcination) f-MWCNTs@Pd/TiO₂ have been deeply investigated. The conversion from amorphous to crystalline TiO₂ takes place upon calcination along with the removal of the organic layers covering the MWCNTs and the Pd nanoparticles. As a consequence of calcination we observed a huge increase in the electrocatalytic performance that we attributed to modification of the TiO₂/MWCNT interface (enhancement of the contact between the two phase, increase of surface state density and apparent doping level), while the accessibility of the Pd NP surface remains almost unchanged. The role of the TiO₂ interface is herein envisaged to template the water access to the active Pd-sites, and due to the hierarchical structure of the nanocomposites, it is in principle hard to establish the contribution to the observed effects of calcination to the sole changes in the TiO₂ crystalline structure.

Referee # 2

In this paper, Giovanni Valenti et al. reported the hierarchical, co-axial arrangement of a Pd/ TiO₂ outer layer on functionalized multi-walled carbon nanotubes (f-MWCNT) and their study for catalytic H₂ evolution. Despite the detailed examination of this system I do not see much significant novelties in their work. First, using carbon nanotubes (MWNT or even SWNT) as electronic mediator for HER reaction have been widely reported in the literature (including the previous works from some of the present authors).

The importance of forming effective interfaces with noble metals/oxide or MoS₂ (as Pt substituent) have also been well noted. The methodologies and

chemistry for immobilization onto MWNT used in this paper have been widely studied.

The authors commented on the replacement of Pt with Pd for this potential application. Although the availability of Pd is slightly higher than Pt, the benefits of replacing Pt with Pd are not great (of a related market and Pd is sometimes more costly than Pt) compared with other non-Pt substituents. I do not think it will make much improvement in this situation.

The authors also reported a more superior performance when the immobilised Pd/TiO₂ on MWNT was subjected to calcination/oxidation than those without. What would happen to the organic interface upon calcination? would it be severely reconstructed to lead to aggregation? There was no metal and TiO₂ particle size analysis before and after the heat treatment or any detailed characterization of the interface. The conventional metal surface area measurements by chemisorption or ECSA by electrochemical techniques may be useful to be included. I have unfortunately found the resolution of the presented images rather low and not informative. Overall, I do not recommend this paper to be published in Nature Commun. due to the lack of novelty. I suggest this work should be submitted to more specialised electrochemical or energy journals.

1) *"The authors commented on the replacement of Pt with Pd for this potential application. Although the availability of Pd is slightly higher than Pt, the benefits of replacing Pt with Pd are not great (of a related market and Pd is sometimes more costly than Pt) compared with other non-Pt substituents. I do not think it will make much improvement in this situation."*

Palladium is a noble metal largely employed in industrial chemistry and environmental catalysis, and its cost-efficiency is not comparable to that of other noble metals and in particular to Pt, Ir, and Rh. As an example, few grams of Pd are present in each car converters today, where Pd based catalyst are state of the art systems with respect to the previous Pt based ones, proving the validity of the concept of Pd replacing Pt. The advantages with respect to Pt, are related to cost [<http://www.platinum.matthey.com/prices/price-charts>] and availability [N.N. Greenwald and A. Earnshaw, *Chemistry of the elements*, 1985, Pergamon press]. As reported by Johnson Matthey price charts updated to July 27th, 2016, Pt's average cost through the last month was 1,069.82 \$/Oz, while Pd averaged at 627.41 \$/Oz (41% less

with respect to Pt). This demonstrates that replacing a Pt-based catalytic process with the corresponding Pd one would be of considerable economic value.

Therefore, we believe that the demonstration of boosting HER effect of the hierarchical and hybrid triple-interface for Pd-NPs, with respect to benchmark systems, is significant and, as addressed by the referee, stimulates further studies, such as the modular design of the electrocatalytic nano-architecture based on base metals.

2) What would happen to the organic interface upon calcination? would it be severely reconstructed to lead to aggregation?

The organic interphase is removed by combustion. As shown by typical thermogravimetric analysis (TGA) profiles of a fresh and calcined sample, the weight loss at around 230 °C, characteristic of the organic moieties and present in the former, disappears completely in the latter. This confirms the successful removal of the organic groups (deriving from the MWCNTs functional groups and from the metal precursors' ligands) (see Figure S1 and discussion paragraph in SI).

In addition, calcination optimizes the contact between MWCNTs and TiO₂. The effect of a thermal annealing on the electronic contact within the nanocomposite is reported in Figure S9, in which the M-S plot change after thermal treatment is highlighted. From the figure, the increase of N_D in f-MWCNTs@Pd/TiO₂ is evident (lower slope of the linear E-dependent region) together with the appearance of a region of extended surface states (at potentials more positive than 0.2 V vs. RHE), that was not observed before calcination.

Noteworthy, TEM evidences and ECSA (Electrochemical Surface Area) analysis of the PdO peak indicate that the thermal treatment is not inducing a severe reconstruction of the electrocatalytic interface with major Pd NP aggregation (ECSA values of 0.0058 cm² and 0.0065 cm² were obtained, before and after thermal annealing at 350 °C, respectively). This discussion has been added within the SI (Paragraph 1.4).

3) There was no metal and TiO₂ particle size analysis before and after the heat treatment or any detailed characterization of the interface. The conventional metal surface area measurements by chemisorption or ECSA by electrochemical techniques may be useful to be included.

In the revised version we added the particles size analysis of the electro-catalysts before and after calcination (see figure S2). In particular, ECSA measurements are instrumental to address the status of the electrocatalytic Pd-sites at the interface. ECSA reveals that the thermal treatment does not alter significantly the Pd accessibility as demonstrated by the Pd

electroactive surfaces of 0.0058 cm² and 0.0065 cm² found before and after the calcination respectively (see also previous point).

4) *I have unfortunately found the resolution of the presented images rather low and not informative.*

We apologies for the inconvenience. In the revised version we edited all the figures to a higher resolution, for optimal visualization, also in the compressed PDF file.

Referee # 3

In this work, the authors have demonstrated an assembly of a 3D-electrocatalytic interface, featuring a hierarchical, co-axial arrangement of a Pd/TiO₂ outer layer on multi-walled carbon nanotubes. The resultant Pd/TiO₂/MWCNT exhibited enhanced HER performance compared with Pd/TiO₂. Overall, it is a nice piece of work and can be considered for publication in Nature Communications after addressing the following points:

1) *In the introduction, authors demonstrate that Pt-based catalysts are plagued by the unfavorable market price. However, Pd is also as expensive as Pt.*

As also discussed above, the actual average price of Pd (627.41 \$/Oz) updated to July 27th, 2016, according to Johnson Matthey price chart (<http://www.platinum.matthey.com/prices/price-charts>) is considerably lower than that of Pt (1.069.82 \$/Oz), demonstrating the considerable economic benefit of replacing Pt with Pd.

2) *f-MWCNTs@Pd/TiO₂ reaches a turnover frequency at zero overpotential, TOF₀, of 9460 H₂ h⁻¹. At zero overpotential, the HER reaction does not take place, why the TOF is so high?*

TOF₀ is obtained from the extrapolation of the experimental TOF determined at higher overpotentials. Our approach, is based on the method presented by Saveant and co-workers (J. Am. Chem. Soc., 134, 11235–11242, 2012): analogously to the concept of exchange current i_0 , the TOF₀ is used to benchmark the efficiency of electrocatalysts at the thermodynamic limit, without the complications associated to the kinetics of the real process.

3) *What is a co-axial TiO₂ outer layer?*

We apologise if the sense of this sentence may not be clear. We simply intended the TiO₂ layer which envelopes the MWCNT throughout its length. We modified the sentence and the meaning should now be more clear (see page 5 main text).

4) In Figure 2, where are the Pd nanoparticles? Could the authors provide the HR-TEM and STEM mapping images? How to confirm the size of Pd nanoparticles?

The lattice spacing values of Pd are too close to those of TiO₂ anatase for being discriminated by HRTEM and FFT analysis, also because of the appreciable broadening of the diffraction spots due to particle size and strain. Therefore, HRTEM does not provide unambiguously identification of Pd nanoparticles in the TiO₂ matrix.

The technique that allows us to identify the Pd/TiO₂ structure is STEM-HAADF, due to the Z (atomic number) dependence of its contrast, and by combining it with EDSX chemical maps.

The new Figure S5 shows a STEM-HAADF image of a f-MWCNTs@Pd/TiO₂, with the naked carbon nanotube in the upper left corner of the picture. Pd nanoparticles can be more easily distinguished from TiO₂ structures due to their higher Z number and thus higher brightness. The STEM-EDXS maps can be used to confirm this result. Figure S5 (b) shows the elemental distribution of Ti (blue) and Pd (green) taken on the same area of Figure S5 (a), demonstrating that the brighter particles correspond to Pd nanoparticles (see discussion added in SI, Paragraph 3).

5) "It is important to note that for both Ox-MWCNTs@Pd/TiO₂ and Tour-MWCNTs@Pd/TiO₂ the coverage with the titania layer is not complete, but zones of bare MWCNTs are clearly observed (Fig. 2d)." Again, the authors shall provide HR-TEM images, where the interfaces between TiO₂ and Pd nanoparticles can be directly identified?

As pointed out in the reply to point 4) above, HRTEM is not the right technique to unambiguously identify Pd nanoparticles in the TiO₂ matrix, since the lattice spacing values of Pd are too close to those of TiO₂ anatase. Only when the Pd nanoparticles lie on portions of bare MWCNTs we are able to clearly identify them via HRTEM. Conversely, the STEM-HAADF technique, combined with EDXS, provides a direct identification of the Pd nanoparticles, as clearly shown in Figure S5.

6) What is the active site if the Pd nanoparticles are embedded within the titania layers?

The active sites are accessible Pd atoms. The porosity of the TiO₂ layer ensures the accessibility of the catalytic site as confirmed by Tafel-slope mechanistic evidence and ECSA determination.

7) "Such defects on the surface of the titania nanocrystals (oxygen vacancies and incompletely coordinated Ti atoms),..... water dissociation and adsorbed hydrogen formation would represent the rate determining step of the HER in the present conditions." Where are the experimental and computational proofs?

Tafel analysis is consistent with water dissociation and adsorbed hydrogen formation as the rate determining step of the electrocatalytic mechanism (see previous discussion). Moreover, the ability of metal hydro(oxy)oxides to promote water dissociation is well established (*Science*, 334, 1256-1260 (2011)). The presence of extended surface states in the TiO₂ layers as evidenced by EIS was interpreted as due to the presence of vacancies at the surface of the metal oxide, and it has been reported that water dissociates at vacancies, hence filling them with hydroxyl (*Surf. Sci. Rep.*, 48, 53-229 (2003)).

8) In SI, the HER mechanism seems not be correct and the first step should be the water dissociation (Volmer step).

We apologise for the inconvenience. In the revised version we change the mechanism accordingly.

9) The enhance HER activity of f-MWCNT@Pd/TiO₂ should originate from the improved conductivity after combining MWCNT. The authors may clarify this point.

The MWCNT concentration plays an important role, due to their electronic properties. MWCNTs are metallic and therefore excellent conductors and electron mediators. They can favour electron transfer processes to the inorganic phase. MWCNT dramatically affects the N_D (electron carrier density), as it is demonstrated by impedance spectroscopy (see Figure S7) and by the comparison between nanostructures with 15% and 20% of MWCNT. In line with the decreased N_D , the nanohybrid with a lower MWCNTs content displayed further decreased electrocatalytic efficiency. One of the advantages of the calcination treatment is to make the MWCNTs/TiO₂ contact tighter, leading to nanohybrids with improved performance. This aspect has been further emphasized in the main text (page 12 and Figure S19).

10) The HER activity of Pd/MWCNT should be provided.

As requested, we have synthesized and characterized the f-MWCNTs@Pd sample. Paragraph 10 in the SI report the comparison between the f-MWCNTs@Pd/TiO₂ and f-MWCNTs@Pd. While the f-MWCNTs@Pd/TiO₂ show a stable HER current, with a 15% decay in 45 hours, in the case of the f-MWCNTs@Pd, the current decays by a similar amount within the first 2 hours. This behaviour highlights the essential role of the TiO₂ layer for increasing the stability of the catalyst (see discussion in the main text, page 12 and the new Figures S21 and S22 in SI).

REVIEWERS' COMMENTS:

Reviewer #1 (Remarks to the Author):

The authors have satisfactorily responded the reviewers' comments and well revised their manuscript. I would like to recommend it to be published in the journal.

Reviewer #3 (Remarks to the Author):

In the revised manuscript, the authors have provided additional experiments to address my concerns. I think that the current work can be acceptable for publication in Nature Communications.